

# Advanced feature selection to study the internationalization strategy of enterprises

Álvaro Herrero[1], Alfredo Jiménez[2] and Roberto Alcalde[3]

[1] Departamento de Ingeniería Informática, Universidad de Burgos, Burgos, Spain
[2] Department of Management, KEDGE Business School, Bordeaux, France
[3] Departamento de Economia y Administración de Empresas, Universidad de Burgos, Burgos, Spain

## ABSTRACT

Firms face an increasingly complex economic and financial environment in which the access to international networks and markets is crucial. To be successful, companies need to understand the role of internationalization determinants such as bilateral psychic distance, experience, etc. Cutting-edge feature selection methods are applied in the present paper and compared to previous results to gain deep knowledge about strategies for Foreign Direct Investment. More precisely, evolutionary feature selection, addressed from the wrapper approach, is applied with two different classifiers as the fitness function: Bagged Trees and Extreme Learning Machines. The proposed intelligent system is validated when applied to real-life data from Spanish Multinational Enterprises (MNEs). These data were extracted from databases belonging to the Spanish Ministry of Industry, Tourism, and Trade. As a result, interesting conclusions are derived about the key features driving to the internationalization of the companies under study. This is the first time that such outcomes are obtained by an intelligent system on internationalization data.

# INTRODUCTION

Many companies nowadays invest and conduct activities in multiple foreign markets. However, a successful internationalization strategy is far from easy in a global environment currently characterized by increasing complexity of networks and interconnections and growing competition (*Johanson & Vahlne, 2009*; *Vahlne, 2013*; *Vahlne & Bhatti, 2019*). For these reasons, international strategy requires accurate and insightful information on the main determinants driving foreign investments to be able to implement the most appropriate decisions. Specifically, one of the first and foremost relevant decisions is the selection of the target market. A carefully crafted international investment operation can go completely wrong if the location is not correct. Accordingly, international business scholars have paid great attention to the study of the determinants of internationalization, notably of Foreign Direct Investment (FDI) operations.

Among the myriad of factors playing a relevant role in the firm's choice of an oversea market, previous studies have highlighted that in addition to firm features such as the industry to which the company belongs, the profitability or the size, and the specific characteristics of the country in terms of macroeconomic figures such as the Gross

Corresponding author
Álvaro Herrero, ahcosio@ubu.es

Domestic Product (GDP), GDP per capita, etc., the concepts of bilateral psychic distance (*Dow & Karunaratna, 2006*; *Johanson & Vahlne, 1977*) and experience (*Clarke, Tamaschke & Liesch, 2013*; *Jiménez et al., 2018*) have a notorious influence. Due to managerial bounded rationality (*Barnard & Simon, 1947*), exploring the various possible configurations of variables that may play a critical impact on internationalization is a complicated task that cannot be performed efficiently. Managers in charge of their firm's internationalization, but also policy-makers aiming to attract higher inflows of foreign investments, need to build on sophisticated tools that can extract more insightful information.

To face this challenging issue, the application of Artificial Intelligence (AI) techniques has been previously proposed (*Herrero & Jiménez, 2019*). A wide variety of AI techniques has been previously applied, ranging from Artificial Neural Networks (*Hsu & Huang, 2018*; *Contreras, Manzanedo & Herrero, 2019*) to Particle Swarm Optimization (*Simić et al., 2019*). In the present paper, a combination of Machine Learning methods is proposed. Differentiating from previous work where unsupervised learning is proposed (*Herrero, Jiménez & Bayraktar, 2019*), in the present paper Feature Selection (FS) (*John, Kohavi & Pfleger, 1994*) is proposed in order to identify the subset of features that best characterize internationalization strategies of companies. To do so, advanced classifiers based on Bagged Decision Trees (BDTs) and Extreme Learning Machines (ELMs), are applied. These supervised-learning methods are used to model the fitness function of a FS schema, where an evolutionary algorithm is applied in order to generate different combinations of features in order to predict the internationalization decision of companies with high accuracy. Furthermore, obtained results are also compared with those from other Machine Learning methods that have been previously applied (*Jiménez & Herrero, 2019*) to the same dataset.

Similar, yet different, solutions comprising genetic FS have also been proposed for problems in other fields such as health (*Salcedo-Sanz et al., 2004*; *Maleki, Zeinali & Niaki, 2021*), Bio-informatics (*Chiesa et al., 2020*) or credit rating (*Jadhav, He & Jenkins, 2018*) among others.

Artificial Intelligence methods have been previously applied to FS (*Saad, 2008*); although they are one of the newest proposals in the field of neural networks, ELMs have been previously applied as classifiers under the frame of evolutionary FS, since the seminal work was published (*Meng-Yao et al., 2012*). FS based on both basic ELM and Optimally Pruned ELM was applied in *Termenon et al. (2013)* and *Chyzhyk, Savio & Graña (2014)*, where the data features were extracted from brain magnetic resonance imaging. In *Termenon et al. (2013)* the ELM-based FS was applied under the frame of an image biomarker identification system for cocaine dependance, while in *Chyzhyk, Savio & Graña (2014)* it was applied to better diagnose patients suffering from Alzheimer's Disease. Results were compared to those obtained by Support Vector Machines, $k$-Nearest Neighbour, Learning Vector Quantization, Relevant Vector Machines, and Dendritic Computing.

In *Xue, Yao & Wu (2018)* a variant of ELMs called Error-Minimized ELM (EM-ELM) is applied to measure the quality of each one of the subsets of features generated by a genetic

algorithm. The proposed FS method is compared to some other Machine Learning methods that do only include one (C4.5) basic decision tree. Furthermore, results are obtained from 10 benchmark datasets, none of them from the economics domain. In *Wang et al. (2018)*, ELMs have been proposed for FS once again, but combined with Particle Swarm Optimization, for regression.

Although the Bootstrap Aggregation (Bagging) of decision trees has been also applied to FS (*Panthong & Srivihok, 2015*), to the best of the authors knowledge, it has never been compared to ELM for this purpose. Thus, going one step forward to the previous work, two advanced FS methods are applied in the present paper to a real-life dataset on company internationalization and their results are compared to those previously obtained by some other FS methods.

The internationalization of companies has been previously researched by Machine Learning methods; in *Rustam, Yaurita & Segovia-Vergas (2018)* a dataset of 595 Spanish firms is analyzed by Support Vector Machines (SVMs) in order to predict the success of internationalization procedures. That is, SVMs are applied in order to differentiate between successful and failed internationalization of manufacturing companies. Differentiating from this previous work, the present paper proposes advanced FS to gain deep knowledge about the key features that are considered by companies in order to invest in a foreign country.

The addressed topic of internationalization is explained in "Literature Review", while the Machine Learning methods proposed and applied are described in "Materials and Methods". Obtained results are compiled and discussed in "Results and Discussion" and the conclusions derived from them are presented in "Conclusions".

## LITERATURE REVIEW

The internationalization of firms is a complex managerial problem in which multiple factors need to be accounted for. As previously mentioned, both company-level and country-level characteristics can have a significant influence. Companies will find a very different environment depending which country invest and, conversely, a given host country will present different opportunities and threats to companies depending on the firms' specific resources and capabilities. Accordingly, both levels, company and country, need not to be overlooked.

Among the various determinants of the location choice of multinational enterprises, two constructs have been recently highlighted by scholars given their significance. Thus, recent studies have shown that experience (*Padmanabhan & Cho, 1999*), at the company-level, and bilateral psychic distance (*Clavel San Emeterio et al., 2018*; *Håkanson et al., 2016*; *Nordman & Tolstoy, 2014*; *Yildiz & Fey, 2016*), at the country one, are particularly important for the majority of Multinational Enterprises (MNEs). Furthermore, international business scholars have called for further attention to the multi-dimensional nature of these constructs, warning against the classic and somewhat simplistic perspective taken in many studies in which a single dimension is analyzed and supposed to capture the full effect (*Dow & Karunaratna, 2006*; *Jiménez et al., 2018*; *Berry, Guillén & Zhou, 2010*; *Pankaj, 2001*; *Puthusserry, Child & Rodrigues, 2014*).

Thus, in the early studies on international trade and investment, distance between countries (home and host) was uniquely conceptualized in terms of geography, building on the so-called "gravity model" (*Tinbergen & Hekscher, 1962*; *Kleinert & Toubal, 2010*). Shortly after, scholars added the effect of cultural distance (*Hofstede, Hofstede & Minkov, 2010*; *Kogut & Singh, 1988*; *Barkema, Bell & Pennings, 1996*). Despite the improvement and success of studies incorporating the effect of cultural distance, recent advances in the field have shown that the true determinant of the location choice is the concept of psychic distance (*Tung & Verbeke, 2010*), which is a broader construct encompassing cultural distance (*Dow & Karunaratna, 2006*). The concept of psychic distance was popularized by the Uppsala School (*Johanson & Vahlne, 1977*, *1990*; *Johanson & Wiedersheim-Paul, 1975*; *Vahlne & Johanson, 2017*; *Nordstrom & Vahlne, 1990*; *Vahlne & Nordström, 1992*) and it is typically defined as "the sum of factors preventing the flow of information from and to the market. Examples are differences in language, education, business practices, culture, and industrial development" (*Johanson & Vahlne, 1977*). *Nordstrom & Vahlne (1990)* further develop the concept by emphasizing learning and understanding the foreign market instead of simply accessing the information. Originally, thus, the emphasis of this literature stream was on the link between great psychic distance and the liability of foreignness, but recent extensions of the model have started to emphasize also how psychic distance also affects the establishment of relationships, and the evolution of other aspects such as R&D, and organizational and strategic change processes (*Johanson & Vahlne, 2009*; *Vahlne & Johanson, 2017*; *Brewer, 2007*). Psychic distance has been shown to be significant for various firm-related outcomes such as FDI location (*Ojala & Tyrväinen, 2009*; *Blomkvist & Drogendijk, 2013*; *Magnani, Zucchella & Floriani, 2018*), subsidiary performance (*Dikova, 2009*), entry mode (*Dow & Larimo, 2009*; *Dow & Ferencikova, 2010*), ownership in acquisitions (*Chikhouni, Edwards & Farashahi, 2017*), innovation (*Azar & Drogendijk, 2014*) or export and trade (*Klein & Roth, 1990*). We present in Table 1 a review of these empirical studies on psychic distance.

As it can be observed from Table 1, all of the works employ traditional, deductive statistical estimation techniques. As *Choudhury, Allen & Endres (2021)* highlight, Machine Learning techniques, drawing on abductive and inductive research, offer a complementary perspective that permits the observation and identification of data patterns that other techniques, such as the classic deductive regressions, can overlook due to their constraints to fit the data into pre-determined models. We precisely aim to adopt such perspective to assess the relevance of diverse firm-level and country-level factors in order to contribute to the study of firm internationalization.

Psychic distance comprises both the individual perceptions of distance of a given individual, shaped by the macro-level factors that form those perceptions (*Dow & Karunaratna, 2006*; *Brewer, 2007*; *Dow & Ferencikova, 2010*; *Ambos, Leicht-Deobald & Leinemann, 2019*; *Bhowmick, 2019*). We follow one of the most influential frameworks of psychic distance proposed in the literature, the one by *Dow & Karunaratna (2006)* published in the leading International Business journal (Journal of International Business Studies), in which six different dimensions (called stimuli) are proposed. Specifically, these

**Table 1 Synthesis of the literature on psychic distance.**

| Author | Year | Sample and estimation technique | Scope of the article |
|---|---|---|---|
| Klein & Roth | 1990 | 477 firms in Canada (multinomial logit model) | The authors analyze the impact of experience and psychic distance as predictors of export decision, differentiating between conditions of high vs low asset specificity |
| Dow & Karunaratna | 2006 | 627 country pairs trade flows among a set of 38 nations (multiple regression model) | The authors develop and test psychic distance stimuli including differences in culture, language, religion, education, and political systems. They find that these measures are better predictor than a composite measure of Hofstede's cultural dimensions |
| Chikhouni, Edwards, & Farashahi | 2007 | 25,440 full and partial acquisitions from 25 countries (Tobit regression) | The authors find that the direction of distance moderates the relationship between distance and ownership in cross-border acquisitions. Besides, they also find significant differences when the acquisition is made by an emerging country multinational compared to when it is made by a developed country one |
| Dikova | 2009 | 208 foreign direct investments made in Central and Eastern Europe (ordinary least-squares regression) | The author obtains empirical evidence supporting a positive relationship between psychic distance and subsidiary performance in the absence of market specific knowledge. However, psychic distance has no effect on subsidiary performance when the firm has prior experience in the region or when it has established the subsidiary with a local partner |
| Dow & Larimo | 2009 | 1,502 investments made by 247 firms in 50 host countries (binary logistic regression) | The authors argue that a more sophisticated conceptualization and operationalization of the concepts of distance and international experience increases the ability to predict entry mode, the lack of which is the reason for ambiguous results in previous research |
| Ojala & Tyrvainen | 2009 | 165 Finnish small and medium firms (stepwise multivariable linear regression) | The authors examine the relevance of cultural/psychic distance, geographical distance, and several aspects related to market size as predictors of the target country preference of SMEs in the software industry |
| Prime, Obadia, & Vida. | 2009 | 8 French manufacturing firms (qualitative study) | The authors critically review the concept of psychic distance and contend that the inconsistent results in previous literature are due to weaknesses in its conceptualization, operationalization, and measurement. Building on their grounded theory-based qualitative study with export managers in French manufacturing companies, the authors propose that psychic distance stimuli should cultural issues (i.e., patterns of thought, behaviors, and language prevailing in the foreign markets) and issues pertaining to the business environment and practices (i.e., relationships with businessmen; the differences in business practices; and the local economic, political, and legal environment) |
| Dow & Ferencikova | 2010 | 154 FDI ventures in Slovakia from 87 potential home countries (logistic regression and multiplevariable linear regression). | In this paper the authors employ psychic distance stimuli to analyze FDI market selection, entry mode choice and performance. The find strong empirical support for a significant effect of psychic distance on both market selection and FDI performance, but the results for entry mode choice are ambiguous |
| Blomkvist & Drogendijk | 2013 | Chinese outward FDI (ordinary least squares regression) | The authors analyze how psychic distance stimuli in language, religion, culture, economic development, political systems, education, plus geographic distance affect Chinese OFDI and find that aggregated psychic distance and certain individual stimuli are significant predictors |
| Azar & Drogendijk | 2014 | 186 export ventures into 23 international markets by Swedish companies (structural equation models) | The authors show that psychic distance has a positive effect on innovation. Firms that perceived a high level of differences in psychically distant markets are more likely to introduce technological and organizational innovations in order to reduce uncertainty. Furthermore, they also find that innovation mediates the relationship between psychic distance and firm performance |

(Continued)

| Table 1 (continued) | | | |
| --- | --- | --- | --- |
| Author | Year | Sample and estimation technique | Scope of the article |
| Puthusserry, Child, & Rodrigues | 2015 | 30 British SMEs and their 30 Indian partner SMEs in international business (qualitative methodology) | The authors investigate inter-partner perceptions of psychic distance between Britain and India, examining different dimensions of psychic distance, their impact and modes of coping with them. They find that culturally embedded psychic distance dimensions tend to have less impact and to be easier to cope with than institutionally embedded dimensions and identify four coping mechanisms |
| Magnani, Zucchella, & Floriani | 2018 | Multiple case study methodology (Italy and Brazil). | The authors analyze the role of firm-specific strategic objectives as determinants of foreign market selection together with objective distance and psychic distance |
| Ambos, Leicht-Deobald, & leinemann | 2019 | 1591 managers located in 25 countries (hierarchical linear modeling) | The authors analyze the formation of psychic distance perception and find that that country-specific international experience, formal education, and the use of common language reduce psychic distance perceptions. In contrast, international experience and overall work experience do not have a significant effect. Besides, they find that individual-level antecedents have lower explanatory level compared to country-level ones |
| Dinner, Kushwaha, & Steenkamp | 2019 | 217 firms based in 19 countries (event study methodology) | The authors investigate the role pf psychic distance when multinational enterprises face foreign marketing crises. They find that the relationship between psychic distance and firm performance during marketing crises has a curvilinear shape and that marketing capabilities moderate this relationship |

authors posit that the individual perceptions of psychic distance are shaped by the country differences in education, industrial development, language, democracy, social system, and religion.

Finally, at the company level, we also rely on recent advances in the literature in which studies have shown that the role of experience is much more complex than initially thought (*Clarke, Tamaschke & Liesch, 2013*). Thus, scholars have shown the great influence of the knowledge that firms can obtain from the experience of other firms (*Jiang, Holburn & Beamish, 2014*). Drawing on Organizational Learning Theory (*Cyert & March, 1963*; *Huber, 1991*; *Levitt & March, 1988*), companies are able to observe the behavior of other companies and obtain valuable information for their own strategy formulation and implementation by learning from best practices and mistakes and establishing collaborations (*Argote, Beckman & Epple, 1990*; *Lieberman & Asaba, 2006*; *Terlaak & Gong, 2008*). Especially when other firms share a key characteristic with the focal company (for example the country of origin or the industry to which they belong (*Jiménez & De la Fuente, 2016*)), their previous actions represent a valuable source of information about expected challenges and opportunities, good and bad practices and networking opportunities (*Terlaak & Gong, 2008*; *Meyer & Nguyen, 2005*).

Overall, a correct internationalization strategy is complicated and elusive given the multitude of factors playing a role and their multi-dimensional nature, which calls for further examination of their particular importance. A finer-grained analysis of the determinants of FDI location by multinational companies can provide insightful information to prospective managers who need to make critical decisions that can determine the success, performance, viability and even survival of their enterprises.

## MATERIALS AND METHODS

The present work aims at obtaining the most relevant features from enterprise-country data that will provide enterprise managers with the information to take decisions on internationalization. In this paper we employ a sample of firms coming from two sources belonging to the Spanish Ministry of Industry, Tourism, and Trade and the website of the Foreign Trade Institute (ICEX) (*ICEX Spain Import and Investments, 2020*). We compiled a sample of independent multinational firms operating in overseas markets by conducting FDI operations. Since small and medium firms have distinct capabilities and face specific challenges in terms of access to funding to internationalize, we focus on large firms and follow the well-established criterion of having 250 employes at least (*Jiménez, 2010*). We also focus on investments before 2007 to prevent distortions in the results due to the impact of the subsequent financial crisis (*Jiménez et al., 2018*).

Following previous studies on the internationalization of Spanish multinationals, we collected the following variables for each foreign subsidiary of the companies in our sample:

- Characteristics from host country such as unemployment, total inward FDI, GDP, growth, and population.
- Bilateral geographic distance as measured in the CEPII database (*Centre d'Études Prospectives et d'Informations Internationales (CEPII), 2020*).
- Psychic distance stimuli between countries. We include all 6 dimensions identified by *Dow & Karunaratna (2006)*: education, industrial development, language, democracy, political ideology, and religion. The first one, education, measures the differences on education enrollment and literacy between the two countries building on data from the United Nations. The second one, industrial development, is the principal component result of ten factors including differences in the consumption of energy, vehicle ownership, employment in agriculture, the number of telephones and televisions, etc. The third one, language, measures the genealogical distance between the dominant languages in the countries and the percentage of population in each country speaking the language of the other. The fourth one, democracy, is based on the similarities in terms of political institutions, civil liberties, and the POLCON and POLITY IV indices. The fifth one, political ideology, measures the differences in the ideologies of the executive powers in each country. Finally, the sixth one, religion, measures the differences in terms of the predominant religion between the countries and the percentage of followers of that religion on the other country. Comprehensive data for all the variables across the majority of countries in the world can be found at (*Dow, 2020*). Similarly, a more detailed description of the procedure to calculate the various psychic distance dimensions can be found at that website and at the seminal paper by *Dow & Karunaratna (2006)*.
- Vicarious experience: Following the previous literature on vicarious experience (*Jiménez & De la Fuente, 2016*; *Jiang, Sui & Cao, 2013*) we employ the total count of other Spanish MNEs present in the host country as our measure of vicarious experience. We distinguish between same-sector vicarious experience (total count of other Spanish

MNEs in the host country belonging to the same sector), different-sector vicarious experience (total count of other Spanish MNEs in the host country belonging to a different sector) and total vicarious experience (the addition of same-sector and different-sector vicarious experience).

- Firm product diversification: we distinguish between three alternatives, namely related product diversification, unrelated product diversification, and single-product firms (*Ramanujam & Varadarajan, 1989*; *Kumar, 2009*).
- Industry: we identify five main groups including manufacturing, food, construction, regulated, and unclassified sectors.
- Other firm characteristics: Return on Equity, number of countries where the firm operates, Number of employes, and whether or not the firm is included in a stock market.

All in all, data from 10,004 samples, compressing 25 features, were collected. Features from countries are:

1. Geographic Distance (Log)
2. Psychic Distance—Education
3. Psychic Distance—Industrial Development
4. Psychic Distance—Language
5. Psychic Distance—Democracy
6. Psychic Distance—Social System
7. Psychic Distance—Religion
8. Unemployment
9. FDI/GDP
10. GDP Growth
11. Population (Log)

Features from companies themselves are:

1. Vicarious Experience
2. Vicarious Experience Same Sector
3. Vicarious Experience Different Sector
4. Manufacturing
5. Food
6. Construction
7. Regulated
8. Financial
9. Employees
10. ROE
11. Stock Market

12. Related Diversification

13. Unrelated Diversification

14. Number of Countries

General statistics about the dataset under study are shown in Table 2.

Obtaining knowledge about decision making regarding the internationalization of companies is a challenging task that perfectly suits Feature Selection (FS). There are mainly two methods from the Machine Learning field that are able to identify the key characteristics of a given dataset. Feature extraction is one of the alternatives, but it is not suitable for the present work as it generates new features from the dataset that are not in the original data. In the present work, the target is to select some of the features from the original dataset as conclusions can be generalized and obtained knowledge applied to other problems (i.e., set of companies). Thus, feature selection is the most appropriate method in the present study.

Hence, some advanced FS proposals are applied in the present research with the aim of identifying the key characteristics that lead to positive or negative internationalization decisions.

In general terms, FS consists of a learning algorithm and an induction algorithm. The learning algorithm chooses certain features (from the original set) upon which it can focus its attention (John, Kohavi & Pfleger, 1994). Only those features identified as the most relevant ones are then selected, while the remaining ones are discarded. Additionally, there is also an induction algorithm that is trained on different combinations of features (from the original dataset) and aimed at minimizing the classification error from the given features. Building on Choudhury, Allen & Endres (2021), supervised Machine Learning is applied in the present work as the induction algorithm.

Under the frame of FS, for every original feature, two different levels of relevance (weak and strong) can be defined (Kohavi & John, 1997). A feature is assigned a strong-relevance level if the error rate obtained by the induction algorithm increases significantly and a weak-relevance level is assigned otherwise. In keeping with this idea, strong-relevance features are to be selected from the internationalization dataset in order to know which ones are the most important ones when taking internationalization decisions.

There are three different ways of coordinating the learning and induction algorithms: embedded, filter and wrapper. The wrapper scheme (Kohavi & John, 1997) has been applied in the present work, being the induction algorithm wrapped by the learning algorithm. That is, the induction algorithm can be considered as a "black box" that is applied to the different combinations of original features that are generated. This perfectly suits the addressed problem because the internationalization decision can be modeled as the target class, being a binary classification problem. As a consequence, well-known and high-performance binary classifiers can be used as induction algorithms. Additionally, the selection of features done by the different induction algorithms can be compared and interesting conclusions from the management perspective can be derived.

**Table 2 Descriptive statistics about the analyzed dataset.**

| Feature | Max | Min | Mean | Std. Dev. |
|---|---|---|---|---|
| Geographic Distance (Log) | 4.29 | 2.83 | 3.59 | 0.37 |
| Psychic Distance—Education | 2.78 | 0.10 | 1.17 | 0.61 |
| Psychic Distance—Industrial Development | 1.34 | 0.00 | 0.59 | 0.34 |
| Psychic Distance—Language | 0.53 | −3.87 | −0.52 | 1.53 |
| Psychic Distance—Democracy | 1.89 | 0.00 | 0.37 | 0.44 |
| Psychic Distance—Social System | 0.67 | 0.00 | 0.36 | 0.23 |
| Psychic Distance—Religion | 1.28 | −1.55 | −0.85 | 0.91 |
| Unemployment | 23.80 | 1.30 | 7.85 | 4.10 |
| FDI/GDP | 20.75 | −11.28 | 3.89 | 4.50 |
| GDP Growth | 10.60 | −3.56 | 4.59 | 2.80 |
| Population (Log) | 9.12 | 5.47 | 7.23 | 0.75 |
| Vicarious Experience | 102.00 | 2.00 | 24.74 | 22.50 |
| Vicarious Experience Same Sector | 38.00 | 0.00 | 6.30 | 7.90 |
| Vicarious Experience Different Sector | 94.00 | 0.00 | 18.44 | 18.05 |
| Manufacturing | 1.00 | 0.00 | 0.37 | 0.48 |
| Food | 1.00 | 0.00 | 0.12 | 0.32 |
| Construction | 1.00 | 0.00 | 0.12 | 0.32 |
| Regulated | 1.00 | 0.00 | 0.08 | 0.27 |
| Financial | 1.00 | 0.00 | 0.09 | 0.28 |
| Employees | 5.21 | 2.30 | 3.33 | 0.65 |
| ROE | 77.50 | −104.45 | 15.09 | 17.15 |
| Stock Market | 1.00 | 0.00 | 0.37 | 0.48 |
| Related Diversification | 1.00 | 0.00 | 0.53 | 0.50 |
| Unrelated Diversification | 1.00 | 0.00 | 0.15 | 0.35 |
| Number of Countries | 89.00 | 1.00 | 11.20 | 12.88 |

According to that, different classifiers have been applied as induction algorithms in order to predict the class of the data. In the present paper, both Bagged Decision Trees and Extreme Learning Machines are applied. Furthermore, results obtained by these two methods are compared to those previously generated (*Jiménez & Herrero, 2019*) by Random Forest (RF) and SVM on the very same dataset.

The classifiers are fed with datasets containing the same data instances as in the original dataset but comprising a reduced number of features. In order to generate different combinations of them, standard Genetic Algorithms (GAs) (*Goldberg, 1989*) have been applied in the present paper. The main reason for choosing such approach is that, when dealing with big datasets, it is a powerful mean of reducing the time for finding near-optimal subsets of features (*Siedlecki & Sklansky, 1989*).

When modeling the problem under this perspective, the different solutions to be considered (selected features, in the present research) are codified as $n$-dimensional binary vectors, being $n$ the total number of features in the original dataset. In a FS problem, a value of 0 is assigned to a given feature if it is not present in the feature subset and 1

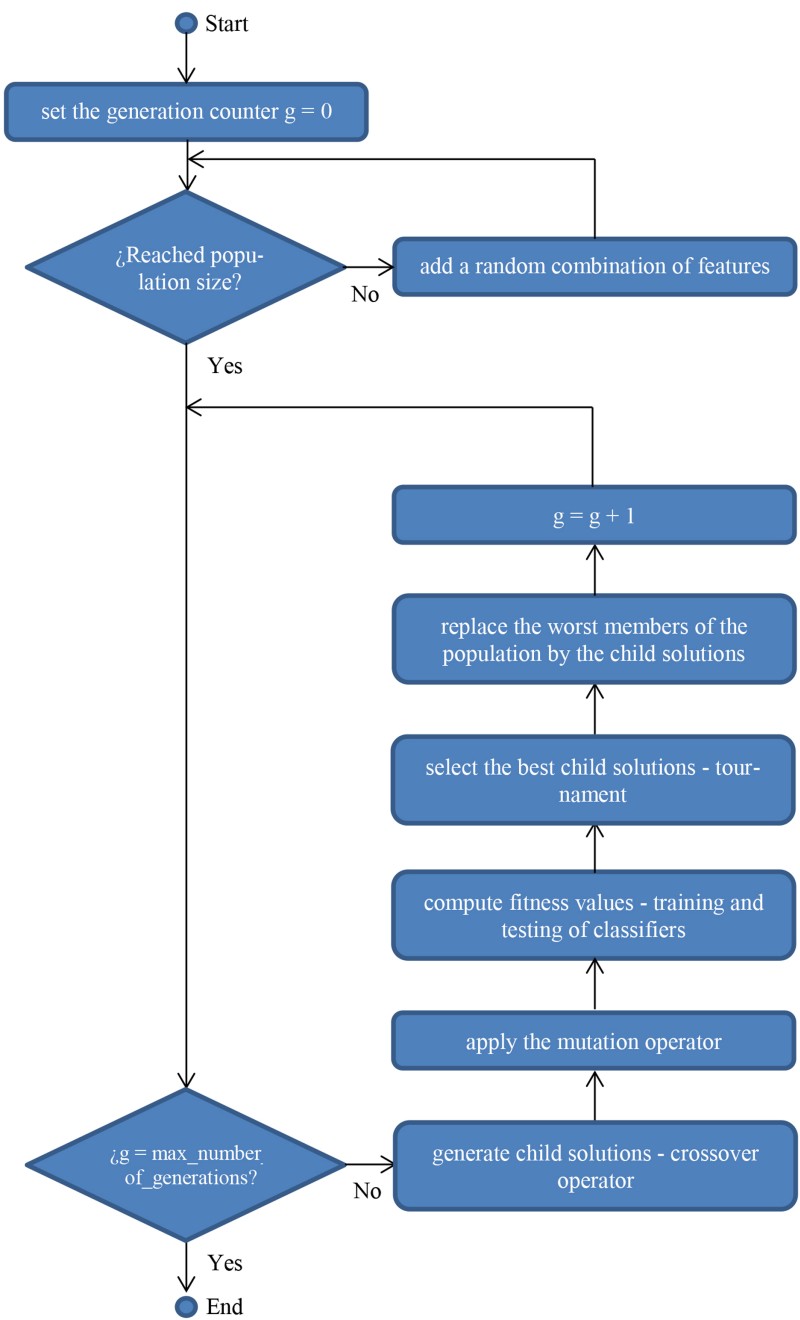

**Figure 1 Flowchart of a standard genetic algorithm for wrapper feature selection.**

otherwise. Once representation of solutions is defined, the GA (as defined in Fig. 1 below) is applied.

In standard GAs, two operators (mutation and crossover) are usually applied depending on a previously stated probability (experimentation has been carried out with different values as explained in "Results and Discussion"). Additionally, when modeling a GA

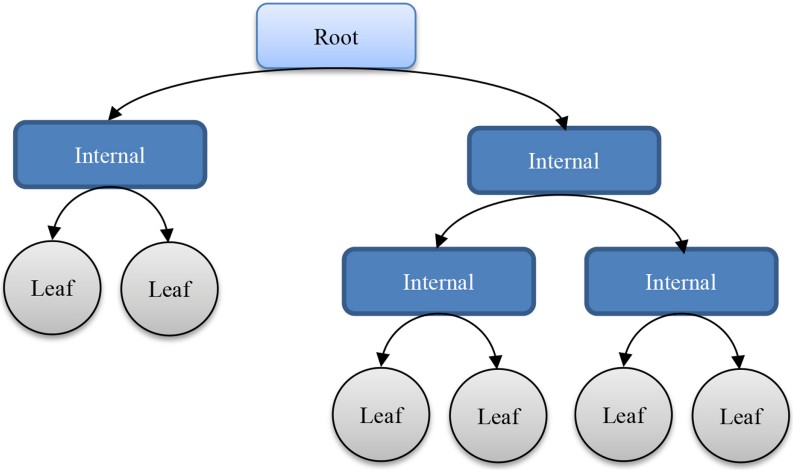

**Figure 2 Sample structure of a decision tree.**

(*Kramer, 2017*), the fitness function is also defined, as the criteria to measure the "goodness" of a given solution, that is its quality.

In the case of FS, the fitness function is usually defined as the misclassification rate of the classifier when applied to the dataset compressing the features in the solution to be evaluated. As a result, the best solution is selected, being the one with the lowest value calculated with the fitness function. That is, the combination of features that led the given classifier to get the lowest error when being tested. As previously mentioned, some different classifiers have been applied, being the novel ones described below.

Decision trees (DTs) (*Safavian & Landgrebe, 1991*) are one of the most popular Machine-Learning methods. They have been successfully applied, proving to be valuable tools for different tasks such as classification, generalization, and description of data (*Sreerama, 1998*). Being trees, they are composed of nodes and arches, as shown in Fig 2.

Nodes in a DT can be of two different types; internal and leaf. The first ones are those aimed at differentiating responses (branches) for a given question. In order to address it, the tree takes into account the original training dataset; more precisely, the values of a certain feature. On the other hand, leaf nodes are associated to the final decision (class prediction) and hence they are assigned a class label. All internal nodes have at least two child nodes and when all of them have two child nodes, the tree is binary. Both parent/child arches and leaf nodes are connected: the first ones are labeled according to the responses to the node question and the second ones according to the classes or the forecast value.

In the present research, performance of DTs is improved by a Booststrap (*Efron & Tibshirani, 1994*) Aggregation (Bagging) strategy, resulting in a Bagged DT (BDT). Within this tree ensemble, every tree is grown on an independently drawn bootstrap subset of the data. Those data that are not included in this training subset are considered to be "out of bag" (OOB) for this tree. The OOB error rate is calculated in order to validate the BDT for each individual (subset of features). For such calculation, training data are not used but those OOB data instances instead.

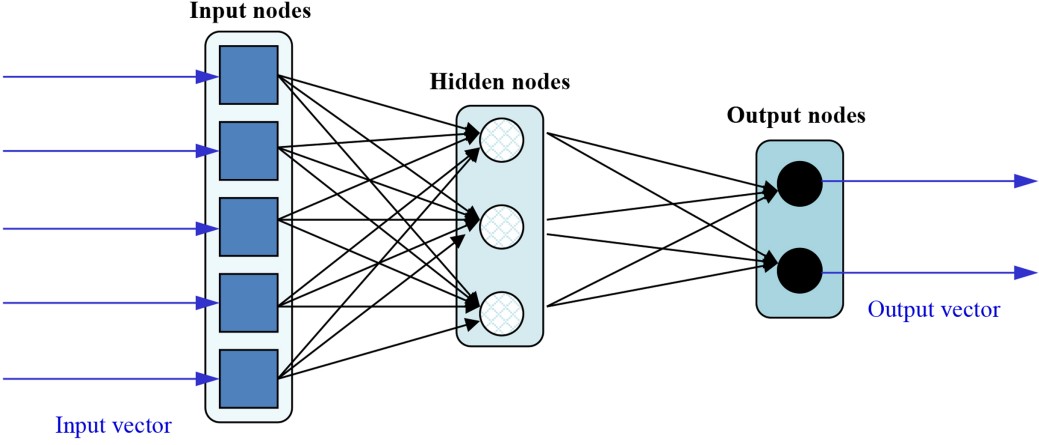

**Figure 3** **Sample topology of an ELM.**

In order to speed up the training of Feedforward Neural Networks (FFNN), Extreme Learning Machines (ELMs) were proposed (*Huang, Zhu & Siew, 2006*). These neural nets can be depicted as:

The functioning of such network can be mathematically expressed as:

$$\sum_{i=1}^{m} \beta_i g_i(x_j) = \sum_{i=1}^{m} \beta_i g(w_i * x_j + b_i) = o_j, \quad j = 1, \ldots, N, \tag{1}$$

being $x_j$ the $j$th input data, $m$ the number of hidden nodes (3 in Fig. 3), $w_i$ the weight vector connecting the input and hidden nodes, $\beta_i$ the weight vector connecting the hidden and output nodes, $b_i$ the bias of the $i$th node, $g_i()$ the activation function of the $i$th hidden node, and $o_j$ the output of the net for the $j$th input data. The ELM training algorithm is pretty simple and consists of the following steps:

- Assign arbitrary input weights ($w_i$) and bias ($b_i$).
- Calculate the hidden layer output by applying the activation function on the weighted product of input values.
- Calculate the output weights ($\beta_i$).

To reduce training time, ELMs are designed as single-hidden layer nets which analytically determine the output weights and randomly choose their hidden nodes. The main consequence of this is that learning speed can be thousands of times faster than traditional FFNN training algorithms like the well-known back-propagation. Unlike the well-known training algorithms based on gradient-based (they try to minimize training error without considering the magnitude of weights), the ELM training algorithm reaches the smallest training error and norm of weights, at the same time. As a side effect, the model also has a good generalization performance and is consequently considered an advisable learning model (for both classification and regression) mainly in those applications when many training processes have to be launched (as in the present case of evolutionary FS).

**Table 3 Parameters values of the GA and misclassification rate associated to the best individual for each classifier.**

| Parameter | Set values | | | |
|---|---|---|---|---|
| | SVM | RF | BDT | ELM |
| Population Size | 30 | 20 | 30 | 30 |
| Number of Generations | 20 | 10 | 20 | 20 |
| Mutation Probability | 0.033 | 0.1 | 0.033 | 0.1 |
| Crossover Probability | 0.9 | 0.9 | 0.9 | 0.6 |
| Misclassification rate | 0.114 | 0.109 | 0.108 | 0.099 |

It is widely acknowledged that one of the main disadvantages of FFNN is that all the parameters need to be tuned. In the case of ELMs, both the hidden layer biases and input weights are randomly assigned, obtaining acceptable error rates.

All the methods previously described in this section have been implemented and run on MATLAB software. For the ELMs the original MATLAB implementation (*Extreme Learning Machines, 2020*) has been adapted to the FS framework.

## RESULTS AND DISCUSSION

As previously explained, a standard GA has been applied to optimize the search of best features subsets. Its most usual parameters were tuned in 81 different combinations, taking the following values:

- Population Size: 10, 20, 30.
- Number of Generations: 10, 15, 20.
- Mutation Probability: 0.033, 0.06, 0.1.
- Crossover Probability: 0.3, 0.6, 0.9.
- Selection Scheme: Tournament.

To get more reliable results, the same combination of values for the parameters above have been used to run the genetic algorithm 10 times (iterations). As stated in "Materials and Methods", the misclassification rate of the classifier has been used as the fitness function to select the best solutions (subset of features). According to the given values of such function, in each experiment the feature subset with the lowest error rate has been selected. Results are shown in this section for the two classifiers that are applied for the first time (BDT and ELM), as well as those for the two previous ones: SVM and RF (*Jiménez & Herrero, 2019*), to ease comparison.

The GA parameter values and the misclassification rate (error) of the best individuals for each one of the classifiers are shown in Table 3. Additionally, for BDTs, 10 trees were built in each iteration and in the case of ELMs, both sigmoidal and sinusoidal functions have been benchmarked as activation functions of the hidden nodes. A varying number of such nodes has been tested as well for each experiment, including 5, 15, 30, 60, 100, 150, and 200 units.

**Table 4 Number of features in the best individuals for the different classifiers.**

| Classifier | Number of features | |
| --- | --- | --- |
| | Best individual | Mean |
| SVM | 11 | 13.1 |
| RF | 17 | 15.7 |
| BDT | 7 | 11.8 |
| ELM | 9 | 8.8 |

In the case of BDTs, the best individual (misclassification rate of 0.108) comprises the following features: "Vicarious Experience Same Sector", "Manufacturing", "Food", "Construction", "Unrelated Diversification", and "Number of Countries". In the case of ELMs, the best individual (misclassification rate of 0.099) can be considered as very robust as it was the best one obtained with both sigmoidal (ELM—sig) and sinusoidal (ELM—sin) output functions and a high number of output neurons (150 and 200 respectively). The one obtained with the sigmoidal function comprises the following ones: "Vicarious Experience Same Sector", "Manufacturing", "Food", "Construction", "Regulated", "Financial", "Employees", "Unrelated Diversification", and "Number of Countries". In the case of the sinusoidal function (ELM—sin), the following features define the best individual: "Vicarious Experience Same Sector", "Manufacturing", "Food", "Construction", "Regulated", "Financial", "Psychic Distance—Language", "ROE", and "Number of Countries".

Best individuals obtained by BDT and ELM share the following features: "Vicarious Experience Same Sector", "Manufacturing", "Food", "Construction", "Financial", and "Number of Countries". When considering the 4 classifiers, the following features are included in all the best individuals "Manufacturing", "Food", and "Number of Countries". On the contrary, "Psychic Distance – Education" and "Unemployment" have not been included in any of the best individuals.

The number of features in the best individuals obtained in the different searches, and the average number of features in the best individuals obtained in all the (10) iterations for the same parameters are shown in Table 4. For further study of the obtained results on advanced feature selection, Fig. 4 shows a boxplot comprising the following information related to the 10 iterations with the combination of parameter values that has generated the best individual in each case. Comprised information includes: mean error, standard deviation error, error of the best individual, and number of features.

From the enterprise management perspective, these results demonstrate the critical importance of vicarious experience, sector, and degree of internationalization as measured by the number of countries where the MNE runs operations. Product diversification, number of employes, and some dimensions of psychic distance are also relevant as they appear in multiple best individuals. However, the results also underline that it is important to disentangle these constructs into their different components, as not all of them are equally important. Thus, the results show that the most critical variable is vicarious experience from other firms in the same sector, but not the one from firms in different

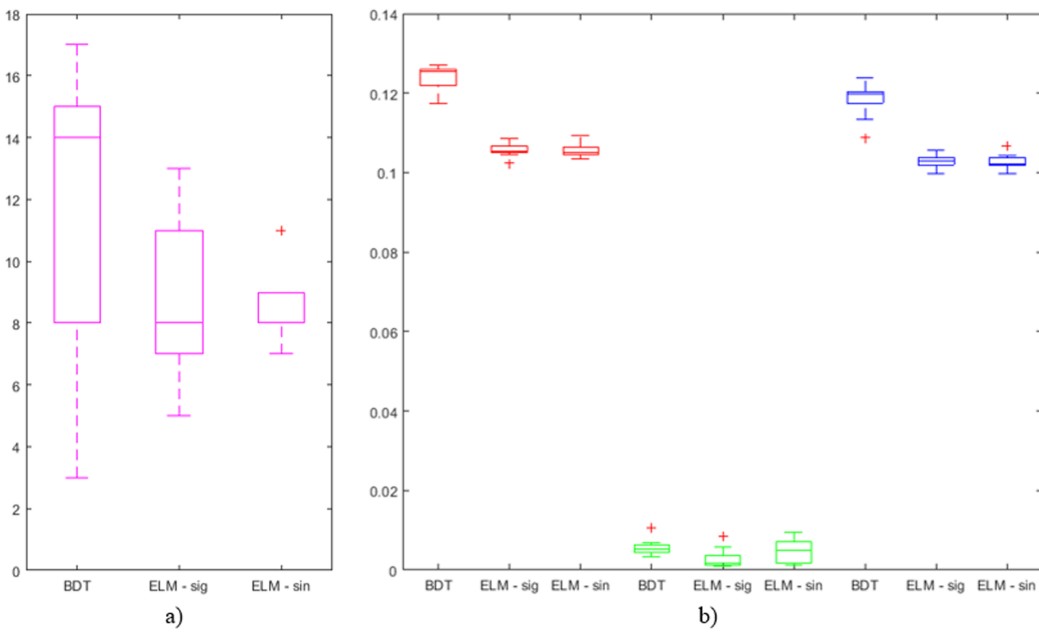

**Figure 4** Boxplots of outputs from iterations on BDT, ELM—sig, and ELM—sin that have obtained the best results: (A) number of features (in magenta) and (B) average error (in red), standard deviation of the error (in green), and error of the best individual (in blue).

sectors. This idea is in line with those of *Jiang, Holburn & Beamish (2014)* where it was shown that firms find vicarious experience from other firms in the same sector much more relevant, valuable, and easier to assimilate. In contrast, vicarious experience from firms in other sectors, while potentially useful (*Jiménez & De la Fuente, 2016*), it is much less applicable as managers will find it more difficult to assimilate and legitimize in front of other stakeholders (*Guillén, 2002*). Similarly, only unrelated diversification appears as relevant, but not related diversification. Further, it is worth noting that only one dimension of psychic distance (i.e., in language) appears as relevant. Among the multiple dimensions, our findings emphasize the importance of communication and the prevention of misunderstandings with other agents in the markets such as customers, suppliers or governments. This result is consistent with recent studies emphasizing the importance of language distance in international business (*Dow, Cuypers & Ertug, 2016*; *Jimenez, Holmqvist & Jimenez, 2019*).

Although the relative lack of relevance of several psychic distance stimuli is somewhat surprising given the amount of studies showing that great psychic distance is detrimental to firms, it is possible that these negative effects are offset by potential positive ones. Some authors have reported that firms devote more resources to research and planning when psychic distance is greater (*Evans & Mavondo, 2002*), whereas when the countries are similar, firms might be complacent and overestimate the similarities, leading to the so-called "psychic distance paradox" (*O'Grady & Lane, 1996*; *Magnusson, Schuster & Taras, 2014*). In fact, various authors consider that firms might take advantage of greater distance as a source of talent and/or knowledge not available in closer markets

(*Nachum, Zaheer & Gross, 2008*) or opportunities for arbitrage, complementarity and creative diversity (*Pankaj, 2001*; *Ghemawat, 2003*; *Shenkar, Luo & Yeheskel, 2008*; *Zaheer, Schomaker & Nachum, 2012*; *Taras et al., 2019*).

Overall, the results clearly depict a complex and multi-dimensional reality in which constructs that are frequently mentioned as determinants of internationalization (experience, distance, and diversification) are indeed complex constructs made of multiple layers that need to be disentangle and analyzed separately to fully understand the impact of each component (*Dow & Karunaratna, 2006*; *Jiménez et al., 2018*; *Berry, Guillén & Zhou, 2010*; *Pankaj, 2001*). Further, the results of the various classifiers consistently point to the critical role of the resources accumulated by the MNE both in terms of employes and own experience in multiple international markets. Finally, these results reinforce the utility of Machine Learning approaches as a complementary tool for researchers, as they permit the identification of patterns from and abductive and an inductive way that other variables employed for deductive causal inference could overlook (*Choudhury, Allen & Endres, 2021*; *Choudhury et al., 2019*).

The main target of the present research has been to reduce the number of features to look for when taking internationalization decisions. It can be easily checked that this objective has been achieved when looking at the number of features comprised in the best individuals. It is worth mentioning the case of BDTs, where only 7 out of the original 25 features (reduction of 72%) were selected while the misclassification rate is the second lowest. In the case of ELMs, it has been reduced up to 9, obtaining the best classification error. Something similar can be said when analyzing the average number of features that is significantly lower than the number of features in the original dataset. The lowest average value (8.8) is taken when applying ELMs and hence for the lowest classification error. When looking at the deviation of the number of features for the best individuals in the 10 iterations (Fig. 4A), it can be said that the mean and the median are quite close in the case of ELMs results, with a significantly small deviation in the case of ELMs with the sinusoidal function (ELM—sin). In the case of BDTs, values greatly vary (from 3 to 17).

Regarding the average error (in red in Fig 4B) of the whole population in the last generation, ELMs have obtained lower values than BDTs. More precisely, the lowest value of all the iterations (0.102) has been obtained by ELM—sig (identified as an outlier in the boxplot). Additionally, the values obtained by ELMs are very compact (low standard deviation). The same can be said about the error of the best individuals (in blue in Fig. 4B). Finally, when analyzing the standard deviation of the error (in green in Fig. 4B), it can be concluded that the highest value (identified as an outlier in the boxplot) has been obtained by the BDT while the lowest ones have been obtained by ELM—sig.

Each one of the features in the original set has been analyzed, from an individual standpoint, for a more comprehensive study. It is shown in Table 5 the percentage of best solutions that includes each one of the original features, for the different classifiers in all the 10 iterations. Additionally, the sum of percentages has also been calculated, and features are ranked, in decreasing order according to it.

**Table 5 Inclusion percentage of original features in the best individuals for all the iterations with the different classifiers.**

| # | Feature name | % | | | | | |
|---|---|---|---|---|---|---|---|
| | | SVM | RF | BDT | ELM | SUM BDT+ELM | SUM TOTAL |
| 25 | Number of Countries | 100 | 70 | 100 | 100 | 200 | 370 |
| 2 | Vicarious Experience Same Sector | 100 | 80 | 80 | 95 | 175 | 355 |
| 4 | Manufacturing | 90 | 70 | 100 | 90 | 190 | 350 |
| 20 | Employees | 80 | 100 | 50 | 80 | 130 | 310 |
| 24 | Unrelated Diversification | 80 | 70 | 80 | 45 | 125 | 275 |
| 5 | Food | 50 | 60 | 50 | 80 | 130 | 240 |
| 6 | Construction | 0 | 90 | 60 | 80 | 140 | 230 |
| 23 | Related Diversification | 80 | 90 | 50 | 0 | 50 | 220 |
| 21 | ROE | 20 | 100 | 60 | 35 | 95 | 215 |
| 9 | Geographic Distance (Log) | 40 | 90 | 60 | 15 | 75 | 205 |
| 10 | Psychic Distance—Education | 60 | 70 | 50 | 25 | 75 | 205 |
| 12 | Psychic Distance—Language | 50 | 60 | 60 | 30 | 90 | 200 |
| 7 | Regulated | 60 | 60 | 30 | 40 | 70 | 190 |
| 18 | GDP Growth | 40 | 70 | 50 | 5 | 55 | 165 |
| 22 | Stock Market | 50 | 70 | 40 | 5 | 45 | 165 |
| 8 | Financial | 10 | 60 | 50 | 40 | 90 | 160 |
| 1 | Vicarious Experience | 70 | 20 | 20 | 40 | 60 | 150 |
| 3 | Vicarious Experience Different Sector | 70 | 40 | 0 | 35 | 35 | 145 |
| 17 | FDI/GDP | 60 | 60 | 10 | 10 | 20 | 140 |
| 15 | Psychic Distance—Religion | 30 | 50 | 40 | 0 | 40 | 120 |
| 16 | Unemployment | 50 | 50 | 20 | 0 | 20 | 120 |
| 13 | Psychic Distance—Democracy | 50 | 40 | 20 | 5 | 25 | 115 |
| 19 | Population (Log) | 20 | 40 | 30 | 20 | 50 | 110 |
| 11 | Psychic Distance—Industrial Development | 30 | 20 | 40 | 5 | 45 | 95 |
| 14 | Psychic Distance—Social System | 20 | 40 | 30 | 0 | 30 | 90 |

The key features (those with the highest inclusion percentage) can be selected from Table 5. According to that, the most important features (highest accumulated inclusion rates, above 340) are (in decreasing order of importance): "Number of Countries", "Vicarious Experience Same Sector", and "Manufacturing". These are also the features with a highest inclusion rate in the case of BDTs and ELMs (SUM BDT+ELM in Table 5). More precisely, "Number of Countries" can be considered as the top feature as it is the one with the highest inclusion rate and was included in all the best individuals obtained by SVMs, BDTs, and ELMs. From Table 5, the least important features (lowest inclusion percentage) can be also identified. They include "Psychic Distance—Democracy", "Population (Log)", "Psychic Distance—Industrial Development", and "Psychic Distance—Social System", that have obtained the lowest accumulated inclusion rates (below 120).

These most and least important features reinforce the ideas discussed above in terms of the relevance of the resources accumulated by the firm in terms of manpower and previous experience in multiple international markets. Also in line with the previous findings, vicarious experience from firms in the same sector and the specific industry to which the MNE belongs to (notably in the case of Manufacturing) manifest themselves as critical determinants. In contrast, other sources of vicarious experience such as the one from firms in other sectors or the combination of vicarious experience from the same and different sectors have a much less important role. As such, the results align more with those found by *Jiang, Holburn & Beamish (2014)* than with *Jiménez & De la Fuente (2016)*. The results also underline the fact that psychic distance do not appear as a critical determinant, and only the dimensions of education and language are moderately relevant. As in the previous case, these results therefore underline the relevance of language distance in international business (*Dow, Cuypers & Ertug, 2016*; *Jimenez, Holmqvist & Jimenez, 2019*), and point to the potential confounding effect of the positive and negative effects of distance in the rest of dimensions (*Pankaj, 2001*; *Ghemawat, 2003*; *Shenkar, Luo & Yeheskel, 2008*; *Zaheer, Schomaker & Nachum, 2012*; *Taras et al., 2019*).

These results (BDTs and ELMs) can be compared with previous ones obtained by SVMs and RFs, and they are consistent. However, they are different in the case of features "Employees" and "Manufacturing". The first one obtained the highest inclusion rate by combining SVMs and RFs while it is the fifth one when considering BDTs and ELMs. Similarly, "Manufacturing" has obtained the second highest inclusion rate by combining BDTs and ELMs. It was identified as the fifth most important one when considering SVMs and RFs. On the other hand, when comparing the least important features, "Psychic Distance—Social System" and "Psychic Distance—Industrial Development" were identified by SVMs and RFs. "Psychic Distance—Democracy", "FDI/GDP", and "Unemployment" have been identified by BDTs and ELMs. From this comparison of classifier results, it can be observed that while all the learning models emphasize the importance of firm-level determinants over country-level ones, SVMs and RFs find the size of the MNE as measured by the number of employes to be more relevant whereas for BDTs and ELMs it has a more modest role and, in contrast, the influences of the industry to which the MNE belongs are more prevalent. Regarding the least important features, all the classifiers identify country-level characteristics, such as various dimensions of psychic distance and some macroeconomic figures related to the economy international openness and the labor market.

## CONCLUSIONS

In this paper we aim to employ sophisticated AI techniques to explore the various possible configurations of variables that may play a critical impact on internationalization, in order to overcome limitations related to bounded rationality (*Barnard & Simon, 1947*) and provide insightful information relevant for managers and policy-makers. From the previously presented results, it can be concluded that advanced FS can be successfully applied in order to identify the most and least relevant features concerning the

internationalization strategy of enterprises. More precisely, the ELM has proved to be the wrapper learning model able to obtain the lowest error when predicting the internationalization decision on the dataset under study.

The results obtained in this research clearly show that firm-level characteristics are more relevant than country-level ones. Perhaps more importantly, the findings underline that constructs such as experience, product diversification or (psychic) distance are indeed complex and multi-dimensional, and that not all their components have the same importance. It is therefore necessary that future works take this complexity into consideration and researchers refrain from employing aggregated measures of these constructs and, instead, test the individual effects of each component or dimension. Overall, the results in fact are consistent with previous works and with the state of the art, but also serve to provide empirical evidence that can contribute to unresolved debates in the literature (i.e., regarding the type of vicarious experience or dimension of psychic distance with the utmost importance). In this sense, we concur with recent research (*Choudhury et al., 2019*; *Choudhury, Allen & Endres, 2021*) highlighting the complementarities between Machine Learning techniques and other traditional tools, as the former permit identifying patterns from an abductive and an inductive way that deductive approaches such as classic regression, due to their constraints to fit models, sometimes overlook.

We acknowledge that our paper is subject to some limitations, which open up interesting opportunities for further research. First, we are unable to include additional variables that could be relevant, such as the percentage of exports or the exact year the company started international operations, due to data unavailability in the data sources we were able to access on Spanish MNEs. Besides, a transnational study is planned as future work, comprising data from additional countries. Additionally, some other classifiers and combinations of them will be applied, trying to get and even lower misclassification rate.

## ACKNOWLEDGEMENTS

The work was conducted during the research stays of Álvaro Herrero and Roberto Alcalde at KEDGE Business School in Bordeaux (France)

### Funding

The authors received no funding for this work.

### Competing Interests

The authors declare that they have no competing interests.

### Author Contributions

- Álvaro Herrero conceived and designed the experiments, performed the experiments, analyzed the data, performed the computation work, prepared figures and/or tables, authored or reviewed drafts of the paper, and approved the final draft.

- Alfredo Jiménez conceived and designed the experiments, performed the experiments, analyzed the data, prepared figures and/or tables, authored or reviewed drafts of the paper, and approved the final draft.
- Roberto Alcalde conceived and designed the experiments, performed the experiments, analyzed the data, authored or reviewed drafts of the paper, and approved the final draft.

## Data Availability

Raw data and code are available in the Supplemental Files.

## Supplemental Information

Supplemental information for this article can be found online at http://dx.doi.org/10.7717/peerj-cs.403#supplemental-information.

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
