# Peer review of "Advanced feature selection to study the internationalization strategy of enterprises"

_PeerJ Computer Science, doi:10.7717/peerj-cs.403_

## Round 0.1 · original submission · Major Revisions

The reviewers agree on the potential of your research.

However, there are serious improvements to be incorporated in order to justify its publication. In order to do so, carefully review their suggestions and if you consider, implement them in your research.

Pay special attention to serious flaws as the need for improvement in the theoretical framework and the methodological aspects that particularly concern reviewer 1 and 2.

Afterwards, if you want to continue with the review process, resubmit the new version of your manuscript together with the letter to reviewers, addressing all comments.

Reviewer 1 ·

Basic reporting

The abstract should clarify aspects such as Design/methodology/approach, Purpose, Findings, Practical implications and above all Originality/value of the manuscript. I don't think the reader understands very well the objectives of the manuscript and the aspects described above.
In the introduction, before embarking into the research objective, the logical stance of the problem statement needs to be fully clarified.
Note that all acronyms/abbreviations must be defined the first time they appear in the abstract, main text, and in figures or tables. For example: MNE.

The section on internationalization of companies should be a section called Literature Review. Authors could incorporate some table of the major papers, their scope, and the techniques used to study the research problem.

For the quality of the literature review, please provide a context for why based on the internationalization of companies.
From line 133 of this last section, I think they should move the whole text to Material & Methods.
I recommend the authors to update the text with the new works of Johanson, J. and Vahlne, J.-E.
Conclusions: Based on these findings, have the current data (yours and others) supported or differentiated? Need to link the results to extant literature, clearly articulate the knowledge gained as results of this study and how this knowledge can be used.

Experimental design

Authors should justify in detail why they have chosen this data analysis technique and justify their choice with a list of similar papers in related research fields such as marketing or business economics.

Validity of the findings

Besides the descriptions of the results of the Hypotheses, more discussion should be offered in more depth.

Additional comments

The authors have produced a well-written and interesting manuscript, but it lacks adequate literature review.
The study of the problem addressed by this manuscript has a wide list of authors and works that are not cited and related in the article.
Although the main strength is the technique of data analysis, it does not adequately justify its choice. The discussion section should be improved, putting the results in relation to other similar research. The conclusions should be improved in depth.

Reviewer 2 ·

Basic reporting

More attention must be paid to the clarity of expression and readability, such as sentence structure, jargon use, acronyms, etc by a professional native speaker. For example in lines 47-48 the authors said: "In present paper, a combination of machine learning methods is proposed in present paper".

There is no sufficient field background/context provided. The literature review section needs to be further developed. On the one hand, from line 133 the variables to be used are described, so all this information, in my opinión, should be in the materials and method section. In this way, the literature review would be limited to only 55 lines, which is in my opinion very scarce.

On the other hand, and more importantly, there are numerous studies on internationalization that have used the concept of psychological distance and that have been published in scientific journals indexed in the Journal Citation Reports or Scimago and that are not included in their review section. from the literature such as: Barkema et al., 1996; Benito and Gripsrud, 1992; Berry et al .; 2010; Brewer, 2007; Clavel et al. 2018; Chikouni et al., 2017; Dikova, 2009; Ghemawat, 2001; Johanson and Wiedersheim-Paul, 1975; Hakanson et al., 2016; Klein and Roth, 1990; Kogut and Singh, 1988; Kostova, 1999; Nordstrom, 1991; Nordstrom and Vahlne, 1994; Padmanabán and Cho, 1999; Prime et al., 2009; Rovira and Tolstoi, 2014; Yildiz and Fey, 2016).

Likewise, the authors make statements without identifying those previous studies, such as (lines 96-99): "Thus, recent studies have shown that experience, at the company-level, and bilateral psychic distance, at the country one, are particularly important for the majority of Multinational Enterprises (MNEs) ". I suggest when the authors mention "recent studies" or "previous studies" include references to those articles.

Finally, with respect to the variables used, a justification is necessary about the reasons for including these variables and not others. For example: In Features from companies themselves, why is the export percentage not included? or why is the year in which they started their international activity not included? Both variables could partly explain their international experience and this would undoubtedly determine, according to influential authors in international literature such as Johanson and Vahlne, the choice of one country or another. To solve this, I suggest a greater and better justification of the variables used is necessary.

Experimental design

Regarding the experimental design: The methods are not described with sufficient information to be reproducible by another investigator. On the one hand, the article does not include how the different variables used have been measured. For example: how did you measure the differences on education enrollment and literacy between the two countries? Or how did you measure Vicarious Experience? The article could be improved if this essential information is included. On the other hand, I would recommend including a table with descriptive statistics that allow the reader to know the characteristics of the sample that has been used in the study.

Validity of the findings

In the results and discussion section, there is little discussion of the results (the authors only include approximately 3 bibliographic references). Most of that section is limited to results and is necessary develop much more the discussion. Thus, I recommend further development in the discussion of results. Additionally, perhaps it might be interesting to separate the results and discussion section into two different sections for clarity to the reader.

Additional comments

The article deals with an interesting topic that can be useful for the scientific and also business fields. However, there are certain issues that have been discussed previously and that in my opinion should be improved before the article is published. The methodology used and the results are interesting, but a more in-depth study of the subject that deals with internationalization is necessary. The main questions for improvement that I suggest are related to the literature review sections, justification for the use of these variables and their way of measuring, and also a greater discussion of the results that they contribute to the scientific and business community.
Best of lucks

---

## Round 0.2 · Minor Revisions

After reviewing your work, reviewers suggest minor improvements. Address everything they comment and submit your work

Reviewer 1 ·

Basic reporting

ok

Experimental design

The authors have made detailed responses to all of the reviewers' comments. However, I believe that the method of analysis provided should be better explained to the reader.
For example, there are important issues that could be added as a list of papers that have used the same methodology. Clarify in more detail the software or tools used for the calculations made.
And adequately justify why it is the most appropriate methodology.

Validity of the findings

ok

Additional comments

If the authors expand on this information, the research work could be ready for definitive acceptance.

Reviewer 2 ·

Basic reporting

Authors should review the journal's publication standards in terms of format or formal aspects. On line 319, for example, the authors include a reference without brackets.

The literature review section has improved. However, I suggest that Table 1 should be further developed including more studies.

Experimental design

No comment

Validity of the findings

The authors have improved the discussion and results section, but should improve the conclusions section in depth. Conclusions should be well stated, linked to original research question & limited to supporting results.

---

## Round 0.3 · accepted · Accept

The current revision satisfactorily addresses the reviewer comments from the last round and I can recommend that the paper be accepted in its current stage.